# Analysis of Post-Transplant Lymphoproliferative Disorder (PTLD) Outcomes with Epstein–Barr Virus (EBV) Assessments—A Single Tertiary Referral Center Experience and Review of Literature

**DOI:** 10.3390/cancers13040899

**Published:** 2021-02-21

**Authors:** Eric Lau, Justin Tyler Moyers, Billy Chen Wang, Il Seok Daniel Jeong, Joanne Lee, Lawrence Liu, Matthew Kim, Rafael Villicana, Bobae Kim, Jasmine Mitchell, Muhammed Omair Kamal, Chien-Shing Chen, Yan Liu, Jun Wang, Richard Chinnock, Huynh Cao

**Affiliations:** 1Department of Medicine, Division of Hematology and Oncology, Loma Linda University Adventist Health Center, Loma Linda University, Loma Linda, CA 92354, USA; justintmoyers@gmail.com (J.T.M.); Mkamal@llu.edu (M.O.K.); Cschen@llu.edu (C.-S.C.); hcao@llu.edu (H.C.); 2Department of Pediatrics, Division of Critical Care Medicine, Loma Linda University Children’s Hospital, Loma Linda, CA 92354, USA; BCWang@llu.edu; 3School of Medicine, Loma Linda University, Loma Linda, CA 92354, USA; is.daniel.jeong@gmail.com (I.S.D.J.); jolee3@students.llu.edu (J.L.); 4Department of Medicine, Washington University, St. Louis, MO 63110, USA; lwliu@wustl.edu; 5Department of Nephrology, University of California, Irvine, CA 92868, USA; matt.sg.kim@gmail.com; 6Department of Medicine, Division of Nephrology, Loma Linda University Adventist Health Center, Loma Linda University, Loma Linda, CA 92354, USA; Rvillicana@llu.edu; 7Department of Medicine, Loma Linda University Adventist Health Center, Loma Linda, CA 92354, USA; BobaeKim@llu.edu (B.K.); JaMitchell@llu.edu (J.M.); 8Department of Pathology, Loma Linda University Adventist Health Center, Loma Linda University, Loma Linda, CA 92354, USA; YanLiu@llu.edu (Y.L.); JWang@llu.edu (J.W.); 9Department of Pediatrics, Loma Linda University Adventist Health Center, Loma Linda University, Loma Linda, CA 92354, USA; Rchinnock@llu.edu

**Keywords:** posttransplant lymphoproliferative disorder, lymphoma, Epstein–Barr virus, solid organ transplant, EBV DNA PCR, infants

## Abstract

**Simple Summary:**

Post-transplant lymphoproliferative disorders (PTLDs) are lymphoid or plasmacytic proliferations that develop in up to 10–15% of immunosuppressed recipients of solid organ transplantation (SOT), bone marrow and/or hematopoietic stem cell allograft. Its prevalence is expected to rise as transplant numbers increase. We performed a single-center retrospective study on the characteristics and outcomes of PTLD at our center in the rituximab era. Infants have been suggested to be a unique population of patients who develop PTLD later due to delayed Epstein–Barr virus (EBV) infection and the presence of maternal antibodies. We found that when compared to older cohorts, infant recipients of SOT had a numerically longer time to PTLD diagnosis. Epstein–Barr virus (EBV) positivity has not been shown to impact survival of patients with PTLD, but we suggest that EBV viral load at time of diagnosis could be investigated further as a marker of survival in patients with EBV-positive PTLD.

**Abstract:**

Post-transplant lymphoproliferative disorders (PTLDs) are lymphoid or plasmacytic proliferations ranging from polyclonal reactive proliferations to overt lymphomas that develop as consequence of immunosuppression in recipients of solid organ transplantation (SOT) or allogeneic bone marrow/hematopoietic stem cell transplantation. Immunosuppression and Epstein–Barr virus (EBV) infection are known risk factors for PTLD. Patients with documented histopathologic diagnosis of primary PTLD at our institution between January 2000 and October 2019 were studied. Sixty-six patients with PTLD following SOT were followed for a median of 9.0 years. The overall median time from transplant to PTLD diagnosis was 5.5 years, with infant transplants showing the longest time to diagnosis at 12.0 years, compared to pediatric and adolescent transplants at 4.0 years and adult transplants at 4.5 years. The median overall survival (OS) was 19.0 years. In the monomorphic diffuse large B-cell (M-DLBCL-PTLD) subtype, median OS was 10.7 years, while median OS for polymorphic subtype was not yet reached. There was no significant difference in OS in patients with M-DLBCL-PTLD stratified by quantitative EBV viral load over and under 100,000 copies/mL at time of diagnosis, although there was a trend towards worse prognosis in those with higher copies.

## 1. Introduction

Post-transplant lymphoproliferative disorders (PTLDs) are lymphoid or plasmacytic proliferations that develop as consequence of immunosuppression in recipients of solid organ transplantation (SOT) or allogeneic bone marrow/hematopoietic stem cell transplantation (BMT/HSCT). The spectrum of PTLDs range from usually Epstein–Barr virus (EBV)-driven polyclonal lymphoid/plasmacytic proliferations to EBV-positive or EBV-negative monoclonal B-cell lymphomas/plasma cell neoplasms or T/NK-cell lymphomas as well as classic Hodgkin lymphoma that is indistinguishable from a subset of lymphomas occurring in immunocompetent individuals. In post-SOT adults, PTLD is seen in up to 10–15% of all SOT recipients, occurring most frequently after multi-organ and intestinal transplantation in 20% of cases followed by the lungs and the heart [1]. Nearly 40,000 transplants occurred in 2019, with transplantation numbers continuing to rise yearly after stagnation in the past decade [2,3]. Optimization of the approach to PTLD is of importance as the prevalence of PTLD is expected to rise as transplant numbers continue to increase [4]. 

The 2017 revised fourth Edition of the World Health Organization classification of Hematopoietic and Lymphoid Tumors classifies PTLD into four categories: Non-destructive PTLD (ND-PTLD, including three subtypes), Polymorphic PTLD (P-PTLD), Monomorphic PTLD (M-PTLD, including B-cell and T-/NK-cell types), and Classic Hodgkin lymphoma PTLD (HL-PTLD) [5,6]. The majority of PTLD is of B-cell origin, with CD20+ monomorphic diffuse large B-cell lymphoma (M-DLBCL) accounting for the majority of cases, whereas 5–10% are T/NK or classic Hodgkin lymphoma-type [1,7]. P-PTLDs are the second most common category of PTLD and make up between 6% and 27% of cases in retrospective series [7,8,9,10]. They are destructive lymphoplasmacytic proliferations that do not fulfill the strict criteria of lymphomas and are difficult to diagnose [11,12].

The prognosis of PTLD has improved following the advent of anti-CD20 monoclonal antibody, rituximab, and lymphoma-specific regimens [1,10,13]. Based mostly on published retrospective evidence, the initial step in the care for all patients with PTLD is reduced immunosuppression (RIS), after which about 60% of patients will need second-line therapy [14,15,16,17]. The landmark prospective multi-center phase 2 PTLD-1 trial set the standard of care for CD20+ PTLD unresponsive to initial RIS. The PTLD-1 trial included seventy patients with PTLD unresponsive to initial RIS, of which 85% were M-PTLD, gave rituximab and stratified those with initial complete response to continued single-agent rituximab and those with inferior response to rituximab with cyclophosphamide, hydroxydaunorubicin, Oncovin, and prednisone (R-CHOP). Complete response was achieved in 70% with a median OS of 6.6 years [18,19]. Because of the poor response of RIS in M-PTLD, some experts suggest administering rituximab in addition to RIS for upfront therapy, and suggest a similar approach in P-PTLD [20]. Although understanding on optimal management of M-PTLD has been improving, there remains a relative paucity of large prospective or randomized trial data regarding the optimal approach for histologic categories that are responsive to upfront RIS, such as ND-PTLD and P-PTLD.

Epstein–Barr Virus (EBV) infection is a major risk factor for the development of PTLD. Several single-center analyses reported that pre-transplant EBV seronegativity can increase the incidence of PTLD by 10- to 76-fold when compared to EBV-seropositive patients due to the risks from primary EBV infection [21,22]. Overall, about 50% of cases of PTLD that develop after SOT are related to EBV, although this has not been associated with differences in response to therapy or survival [1,23]. In patients with EBV-positive PTLD, EBV DNA levels may have a role as a tumor marker, as it is known be released from EBV-positive malignancies during apoptosis or necrosis and was shown to decline in patients who respond to therapy [24,25,26,27]. A prospective study has demonstrated that changes in plasma EBV DNA levels correlated with radiographic tumor response to therapy in EBV-positive malignancies [28]. In another study, the median plasma copies/mL of untreated biopsy proven EBV-positive PTLD was 54,960 and ranged from 170 to 961,520 copies/mL, indicating a wide range at time of diagnosis [27]. The association of EBV viral load in EBV-positive PTLD at time of diagnosis and survival has not been well characterized.

Infants (under one year of age) may represent a unique subset of patients that develop PTLD, potentially owing to unique interactions with immunosuppression and EBV infection. For example, the incidence of PTLD is higher in pediatrics and adolescents (those under 18 years of age) than in adults due to primary EBV infection, and ranges from 1.2% to 30% depending on the transplanted organ [1,29,30,31,32]. However, infants have a lower incidence of PTLD than older pediatric patients [33]. Reasons for this finding are not fully elucidated. One proposed mechanism is that maternal IgG antibodies are known to transfer before birth transplacentally and are protective for bacterial and viral infections [34]. Furthermore, neonates (those under 1 month of age after birth) are relatively more tolerant of allografts and require less immunosuppression [35]. Lastly, infants may have a longer time to primary EBV infection than older patients, resulting in a delayed time to development of PTLD [1,35].

Loma Linda University Medical Center (LLUMC) has been performing SOT since 1967. The first neonatal recipient of heart transplant worldwide with long-term survival occurred at LLUMC in 1985, and our center has extensive experience with heart transplant in infants [35,36]. Here we report our single-center experience in the rituximab era with the clinical characteristics and disease outcomes in different subtypes of PTLD.

## 2. Materials and Methods

Patients with documented histopathologic diagnosis of primary PTLD or those who were treated for PTLD after SOT at Loma Linda University Medical Center (LLUMC) between January 2000 and October 2019 were included. Patients were excluded if they initially had more than one primary PTLD diagnosis. This retrospective study was approved by the LLUMC Institutional Review Board (IRB Approval # 5180348 and # 53306). Due to the retrospective nature of the study with minimal risk to the research patients, the IRB waived informed consent and Health Insurance Portability and Accountability Act (HIPAA) authorization.

Patients with PTLD diagnosis were identified by query of electronic medical record for the diagnosis of PTLD. Clinical characteristics were obtained including age at diagnosis, time from SOT to diagnosis, treatment regimen and response, vital status, and date of last contact or death. Details concerning SOT were obtained including organ transplanted, and date of transplantation. Histologic subcategories and Epstsein-Barr virus-encoded small RNA (EBER) positivity were documented from pathology reports and re-reviewed and confirmed by two expert hematopathologists based on criteria of the newest 2017 revision of WHO classification. Laboratory data and clinical notes were reviewed for whole blood EBV DNA viral loads (EBV-VL) obtained via PCR at time of diagnosis before initiating treatment. Time to progression was calculated from date of histologic diagnosis to date of histologic progression or progression as stated by treating physician.

Transplantation age groups were defined as neonatal (birth to 1 month), infant (birth to 1 year), pediatric/adolescent (1 year to 18 years), and adult (18 years and older) at the time of transplant. Time to PTLD diagnosis was defined as the time between the date of transplant and the date of biopsy confirming histologic diagnosis of PTLD, with early (within the first year of transplant), late (between the first and the tenth year), and very late (after the tenth year). Multi-organ transplant was defined as receiving more than one SOT. The Revised International Prognostic Index (R-IPI) was calculated from available data and was considered low if 0–2 and high if 3–4. Treatment response was determined by clinician-assessed response as complete response (CR), partial response (PR), or progressive disease (PD). Objective response (OR) was defined as the combination of PR and CR. Due to the limited number of patients, we examined treatment regimens by treatment response only in P-PTLD and M-DLBCL-PTLD. Survival time was calculated from the date of positive histologic diagnosis to the date of death or last contact.

Kaplan–Meier survival estimates were used to determine primary outcomes of OS and compared by log-rank method. A *p* < 0.05 was used for statistical significance. Statistical analysis was performed using SPSS Statistics Build Version 26.

## 3. Results

### 3.1. Clinical Characteristics of Total Cohort

66 patients were identified with a histologically-confirmed diagnosis of PTLD (41 male, 25 female) following SOT (38 heart, 21 kidney, 4 liver, and 3 multi-organ). The median follow-up time was 9.0 years with the range between 0 and 24.7 years. Notably, 21/23 (91%) of the grafts received in infants were heart transplants. Clinical characteristics of the study population are summarized in Table 1.

### 3.2. PTLD Histologic Subtype Profile

The majority of patients had the M-DLBCL-PTLD subtype (*n* = 36) with the next largest subset being P-PTLD (*n* = 13), classical HL-PTLD (*n* = 3), and ND-PTLD (4 with infectious mononucleosis-like histology). Four of the 36 cases of M-DLBCL were notably Hodgkin-Like PTLD, which have been re-categorized as M-DLBCL since 2008 by WHO due to having similar clinical and pathologic characteristics [37]. Within M-PTLD, there were 4 Burkitt lymphoma (M-BL-PTLD), 1 T/NK-cell type lymphoma NOS, 1 angioimmunoblastic T-cell lymphoma, 1 anaplastic large cell lymphoma, and 3 plasma cell neoplasms (M-PCN). Patient characteristics are summarized by histologic subtype in Table 2. 

### 3.3. Time to PTLD Diagnosis by Age and Organ Transplanted

The overall median time from transplant to PTLD diagnosis was 5.5 years (95% confidence interval (CI): 2.9, 8.1). Amongst subgroups, infant transplant recipients showed the longest time to diagnosis at 12.0 years (*n* = 23) (95% CI: 8.1, 15.9) compared to pediatric/adolescent transplants at 4.0 years (*n* = 22) (95% CI: 2.1, 5.9) and adult transplants at 4.5 years (*n* = 21) (95% CI: 2.9, 8.1) (*p* = 0.125). Cumulative incidence of these diagnoses is shown in Figure 1A. After restricting this analysis to heart transplant recipients, the difference in time to PTLD diagnosis stratified by transplant age approached, but did not reach statistical significance (*p* = 0.051; Appendix A).

Comparing time to PTLD diagnosis by solid-organ transplanted, the median time to diagnosis of PTLD for liver transplant was at 0.49 years (*n* = 4) (95% CI: 0, 3.5), multi-organ at 2.4 years (*n* = 3) (95% CI: 0.21, 4.6), kidney at 4.0 years (*n* = 21) (95% CI: 1.4, 6.6), and heart transplant at 9.1 years (*n* = 38) (95% CI: 6.2, 12.0) (*p* = 0.525) (Figure 1B). The median time to diagnosis for heart transplant after excluding the infant group was 4.9 years (*n* = 17) (95% CI: 3.5, 6.3).

### 3.4. First-Line Treatment Regimen and Treatment Response

The overall objective response rate (ORR) in the M-DLBCL-PTLD subtype was 72% (*n* = 26); 67% (*n* = 24) achieved CR. Twenty-four patients received rituximab and chemotherapy (R-CTX) regimens as the first-line: R-CHOP (*n* = 12), rituximab, cyclophosphamide, and prednisone (R-CP) (*n* = 10), and one patient each with rituximab, etoposide, Oncovin, cyclophosphamide, and hydroxydaunorubicin (R-EPOCH), rituximab-cyclophosphamide, vincristine, prednisone (R-CVP), and vincristine, cyclophosphamide, rituximab (VCR). Three (13%) patients on R-CTX had PD, all of whom went on to receive second-line therapy.

In the P-PTLD cohort, the ORR was 92% (*n* = 12); all were CR. Five of the P-PTLD patients who achieved CR received RIS with or without AV alone without other interventions. No patients with P-PTLD required subsequent treatment. During treatment, 15/65 (23%) experienced acute rejection with 6/65 (9.2%) graft losses (one unknown). 

Treatment regimens and corresponding treatment responses for M-DLBCL-PTLD and P-PTLD subtypes are found in Table 3. For treatment response stratified by patient age, see Appendix A.

### 3.5. Overall Survival by Age at PTLD Diagnosis and R-IPI

The median overall survival (OS) of our study population was 19.0 years (*n* = 66) (95% CI: 12.5, 25.7) (Figure 2A). Examining age categories at time of PTLD diagnosis showed that the median OS for the infant group was 15.1 years (95% CI: 9.95, not yet reached (NYR)), while the pediatric/adolescent group was NYR with 5-year survival rate of 68.2%, and the adult group was 10.7 years (95% CI: 0, 33.0) (*p* = 0.201). Survival analysis stratified by R-IPI showed the median OS of those in the low score cohort was NYR with the five-year survival rate of 73.3%, whereas those in the high score group was 0.71 years (95% CI: 0, 6.9) (*p* < 0.0001) (Figure 2B).

The median OS for those with the M-DLBCL-PTLD subtype was 10.7 years (95% CI: 5.4, 16.0), whereas the median OS was NYR for those with P-PTLD, with 5-year survival rates of 44% (16/36) and 92.3% (*n* = 12/13), respectively. The OS differences reached statistical significance for that between P-PTLD and M-DLBCL-PTLD (*p* = 0.022) (Figure 3), but differences between other subtypes did not reach statistical significance. The five-year survival rates of other subtypes in our cohort were as follows: M-BL-PTLD: 100% (*n* =4/4), HL-PTLD: 100% (*n* = 3/3), ND-PTLD: 75% (*n* = 3/4), M-PCN-PTLD: 33% (*n* = 1/3), T/NK-PTLD: 0% (*n* = 0/3).

### 3.6. Qualitative EBER and EBV DNA Status

EBER-ISH was performed in 63 of 66 diagnoses. EBER-positive histology was observed in the majority of patients (*n* = 49/63, 78%). EBER positivity was as follows: M-DLBCL-PTLD (*n* = 24/35, 69%) M-PCN-PTLD (*n* = 1, 33%), HL-PTLD (*n* = 2/2, 100%), P-PTLD (*n* = 12/12, 100%), M-BL-PTLD (*n* = 3/4, 75%), ND-PTLD (*n* = 4/4, 100%), and T/NK-PTLD (*n* = 3/3, 100%).

Plasma EBV DNA results were obtained via PCR in 56 of 66 diagnoses. Overall plasma EBV DNA positivity was 89% (*n* = 50/56). Subtype analysis in our cohort revealed positive plasma EBV DNA in all cases of HL-PTLD (*n* = 2/2 100%) and M-BL-PTLD cases (*n* = 3/3, 100%), and most cases of P-PTLD (92%, *n* = 11/12), M-DLBCL-PTLD (*n* = 27/31, 87%), and ND-PTLD (*n* = 3/3, 100%). 

Discordance was observed between tumor EBER status and whole blood EBV DNA. There were two EBER-positive cases that were EBV DNA-negative: one case of M-DLBCL-PTLD and one case of P-PTLD. There were six EBER-negative cases that showed EBV PCR positivity, all belonging to the M-DLBCL-PTLD subtype. EBV and EBER characteristics are summarized by histologic subtype in Table 2.

### 3.7. Quantitative EBV Viral Load

Measurement of whole blood EBV-viral load (VL) at PTLD diagnosis was documented in 56 of 66 patients. Three EBV DNA-positive patients with M-DLBCL-PTLD were missing quantitative EBV-VL. Because the largest subset of patients was those with M-DLBCL-PTLD with 24 positive patients, EBV-VL was examined at 10,000 and 100,000 copies/mL in this subtype. The median pre-treatment EBV-VL in the M-DLBCL-PTLD cohort was 4394 copies/mL, ranging between 21 and 4,063,364. With the 10,000 copies/mL threshold, the median OS for greater than 10,000 copies/mL was 9.0 years (*n* = 9, (95% CI: 0.35, 17.7)) while OS for less than or equal to 10,000 copies/mL was 15.1 years (*n* = 15, (95% CI: 0, 39.9)) (*p* = 0.377) (Figure 4A). The median progression free survival (PFS) of those greater than 10,000 copies/mL was 0.73 years (95% CI: 0.16, 1.31) and less than or equal to 10,000 copies/mL was 10.4 years (95% CI: 0, 29.3) (*p* = 0.186).

With the 100,000 copies/mL threshold, median OS for greater than 100,000 copies/mL was 9.0 years (*n* = 5, (95% CI: 0, 22.3)), while that less than or equal to 100,000 copies/mL was 15.1 years (*n* = 19) (*p* = 0.102) (Figure 4B). The median progression free survival (PFS) of those greater than 100,000 copies/mL was 0.5 years (95% CI: 0.26, 0.74) and less than or equal to 100,000 copies/mL was 10.4 years (95% CI: 0, 23.5) (*p* = 0.174). Clinical characteristics are presented in Appendix A.

When the four cases of Hodgkin-Like PTLD that were re-categorized as M-DLBCL-PTLD were excluded from OS analysis at the 100,000 copies/mL threshold, the median OS for those with greater than 100,000 copies/mL was 9.0 years (*n* = 5, (95% CI: 0, 22.3)), while those with less than or equal to 100,000 copies/mL was 15.1 years (*n* = 16) (*p* = 0.038). The median progression free survival (PFS) of those greater than 100,000 copies/mL was 0.5 years (95% CI: 0.26, 0.74) and less than or equal to 100,000 copies/mL was 12.7 years (95% CI: 4.2, 21.1) (*p* = 0.086).

## 4. Discussion

In this single tertiary center retrospective study, we identified patients who developed PTLD after receiving SOT and examined multiple histologic and disease characteristics.

Compared to pediatric/adolescent and adult cohorts, infant SOT recipients, the majority of whom were heart transplants, had the longest numerical time to PTLD diagnosis (median time of 12.0 years). When this analysis was restricted to the heart transplant cohort, the difference in time to develop PTLD stratified by age at diagnosis approached statistical significance (*p* = 0.051). Overall, only 3/24 (12%) transplanted infants developed PTLD within the first five years, which is consistent with the lower rate of PTLD in infants when compared to older pediatric and adolescent patients [33]. Infants may have delayed time to PTLD diagnosis because of unique interactions with the two major risk factors for PTLD: immunosuppression and EBV infection. An outcomes study of neonatal recipients of heart transplant at our center demonstrated lower rates of rejection than those transplanted at older ages, and this has been hypothesized to be due to increased allograft tolerance in newborns [35]. For this reason, neonatal heart transplants at our center do not undergo induction therapy and less than 5% of infant transplants have historically required maintenance steroid therapy [35]. Furthermore, infants are known to often have EBV IgG antibodies at birth which subsequently decay. This finding is thought to reflect the transplacental transfer of antibodies that provide passive immunity [33,34]. The combination of less immunosuppression and protective maternal antibodies may lower the incidence of early PTLD in the youngest transplant recipients [33,38]. Second, because EBV infection in seronegative patients is a major risk factor for development of PTLD, the longer time to PTLD diagnosis could be related to the timing of EBV infection [1]. National Health and Nutrition Examination Surveys (NHANES) data suggest a gradually increasing prevalence of EBV antibodies at around 50% between the ages of 6–8 and 90% at the age of 18–19 [39]. In accordance with this data, our center found that heart transplant recipients under one year of age had a median time to EBV infection of approximately 7 years [35]. A study with a larger cohort should be conducted to confirm this finding as there may be implications on optimal screening methods for PTLD in infant recipients of SOT.

Heart transplantation also had a numerically longer time from transplant to PTLD diagnosis when compared with other SOTs (median time of 9.1 years). The median time to PTLD diagnosis was 4.9 years after excluding infants, which made up 21/38 (55%) of the overall heart transplant cohort, suggesting the longer time to PTLD in heart transplant is confounded by infants. However, the German Ped-PTLD registry also demonstrated a significantly longer time to PTLD in pediatric heart transplant recipients when compared to other transplants. In this study, liver and heart recipients had a similar age at transplantation and late PTLD was rarely detected in liver recipients. In contrast, patients with heart transplant continued to be at risk for late PTLD and even very late PTLD. This finding was thought to be due to the necessity for long-term immunosuppression in this cohort, whereas immunosuppression can be tapered or discontinued in some liver transplant recipients [29]. Other large database studies similarly observe that while the risk of PTLD in non-heart SOT recipients level off, the risk of PTLD in heart transplant recipients continues to increase over time. For this reason, the median time to PTLD diagnosis is partially dependent on length of study follow-up [33,40]. The relatively long median time of follow-up in our study of 9.0 years that ranged up to 24.7 years allowed us to capture very late cases of PTLD in heart transplant patients. 

We found that nearly all patients with P-PTLD achieved a CR (12/13, 92%), in which five (5/13, 38%) patients received upfront RIS with or without AV therapy without need for subsequent treatment. This was matched by superior OS compared to other histologic categories. Other retrospective studies also suggest high responses to RIS alone in P-PTLD, with 77–100% of patients able to achieve CR [16,41,42]. The National Cancer Comprehensive Network (NCCN) guidelines on P-PTLD have recommended the use of RIS if possible for all cases, and the use of rituximab therapy or chemoimmunotherapy for systemic P-PTLD, versus rituximab or definitive local management for non-systemic P-PTLD [43]. Some experts have recommended the use of upfront combined rituximab with RIS for most patients with CD20+ P-PTLD [20]. However, the use of rituximab up front may not be indicated in all patients with P-PTLD. One recent single-center study compared the use of RIS alone, rituximab (with or without RIS), and chemotherapy (with or without RIS, and with or without rituximab). Surprisingly, they demonstrated significantly poorer survival in patients who had upfront rituximab with or without RIS in both the total cohort and the early and P-PTLD subgroups, despite having comparable performance status and disease status to the rest of the cohort [7]. Choosing which upfront treatment strategy in P-PTLD may depend on patient factors, as some patients are able to achieve good outcomes with RIS alone. One study demonstrated three independent factors predictive of failure to initial RIS in PTLD: age over 50, stage 3–4 disease, or bulky disease, with response rates of 77% with 0 risk factors, 54% with 1 risk factor, and 15% with 2–3 adverse factors [16]. The optimal management of P-PTLD remains unclear, and our observations support the need for a prospective clinical trial to address this area of need.

Our study did not demonstrate a statistically significant relationship of high whole blood EBV-VL at time of diagnosis of M-DLBCL-PTLD with OS. However, in the overall cohort of M-DLBCL-PTLD, there was a trend towards improved OS between those with viral copies over and under 100,000 copies/mL. When comparing patients above versus below this threshold, a larger proportion of patients had higher R-IPI scores: 60% vs. 32%, respectively. Notably, EBV status of PTLD has not been shown to impact response to treatment or survival outcomes in both large retrospective studies and the prospective PTLD-1 trial [23,44,45]. However, these studies did not stratify by EBV-VL in patients with EBV-positive PTLD. Circulating EBV DNA, especially plasma cell free EBV DNA, may be a useful surrogate tumor marker as it is shed during active necrosis or apoptosis and decreases in response to therapeutic interventions [24,27,46]. A prospective pilot study has demonstrated that changes in plasma EBV DNA correlated with radiographic tumor response to therapy in EBV-positive malignancies, including PTLD [28]. In a retrospective study, the median plasma EBV viral copies/mL of untreated biopsy-proven EBV-positive PTLD was 54,960, and ranged from 170 to 961,520 copies/mL, indicating significant heterogeneity at time of diagnosis [27]. We suggest that a larger study should be done with multivariate statistics to investigate if the degree of elevation viral copies of EBV DNA at time of diagnosis is an independent prognostic factor in patients with EBV-positive PTLD.

This is a retrospective single-center study, which limits the significance of our findings and puts our study at risk of selection bias. Although there are plausible mechanisms for infants to have delayed PTLD, this finding did not reach statistical significance due to small cohort size. Furthermore, this finding may also be affected by selection bias, as adults who receive SOT are more likely to have comorbidities and may die before development of PTLD. The number of patients were low for multiple cohorts, including liver and multi-organ transplants, and all subtypes of PTLD, except M-DLBCL-PTLD. As a result, definitive conclusions regarding these smaller cohorts cannot be drawn. Donor and recipient EBV serology status was not available, nor was the time to EBV infection.

## 5. Conclusions

In this single-center retrospective study, we suggest that infants may have a longer time to PTLD diagnosis than older cohorts. Although the interaction p-value for age category as a factor in time to PTLD diagnosis did not reach statistical significance, there is a strong biologic rationale for infants to have a delayed time to PTLD. This should be investigated in a larger cohort due to potential implications for the optimal duration of PTLD surveillance in this population. Although our cohort was limited, we found that P-PTLD patients had a good response to treatment regardless of strategy used, and a notable portion achieved CR with RIS with or without AV alone. We suggest that RIS without additional therapy may be a reasonable strategy in patients with P-PTLD and that this should be investigated further. Lastly, we observed that a high (>100,000 copies/mL) whole blood EBV DNA VL at diagnosis trended towards worse OS, although this did not reach statistical significance. Further studies will be needed to identify if pre-treatment EBV DNA is an independent prognostic marker in those with EBV-positive PTLD.

## Figures and Tables

**Figure 1 cancers-13-00899-f001:**
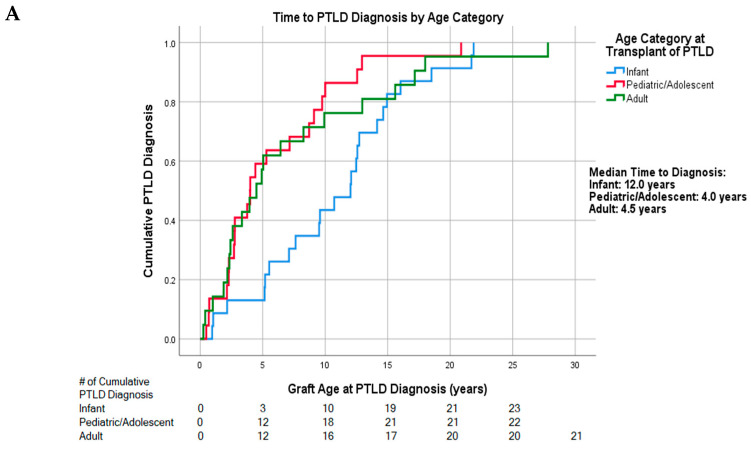
(**A**). Cumulative PTLD diagnosis categorized by age group. (**B**). Cumulative PTLD diagnosis categorized by transplanted organ.

**Figure 2 cancers-13-00899-f002:**
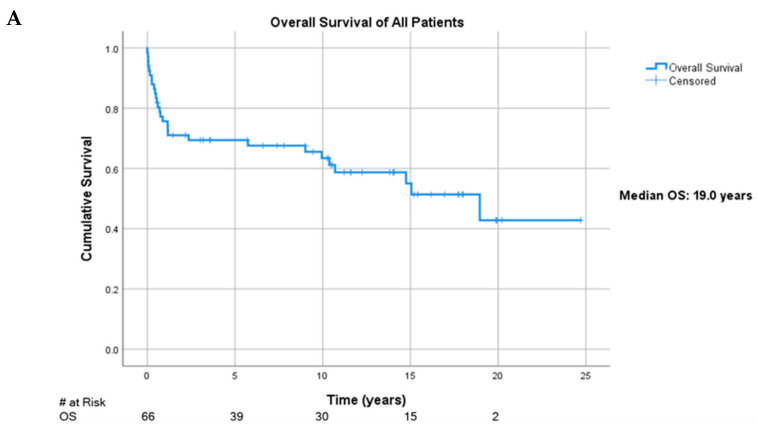
(**A**). Overall survival in this study population. (**B**). Overall survival categorized by Revised International Prognostic Index (R-IPI) staging.

**Figure 3 cancers-13-00899-f003:**
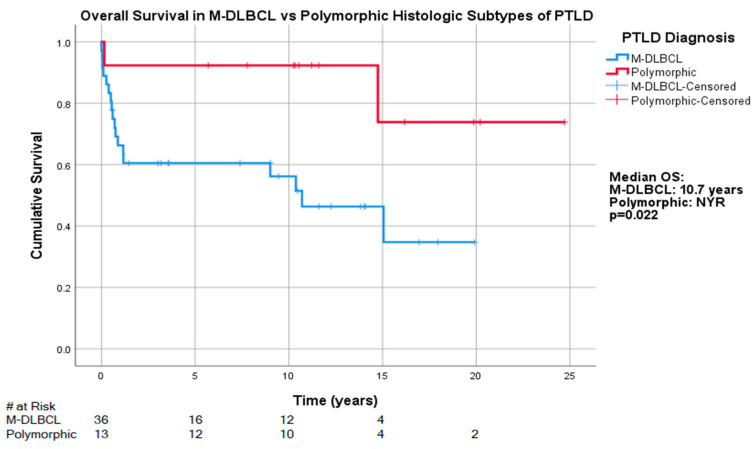
Overall survival in M-DLBCL-PTLD and P-PTLD subtypes.

**Figure 4 cancers-13-00899-f004:**
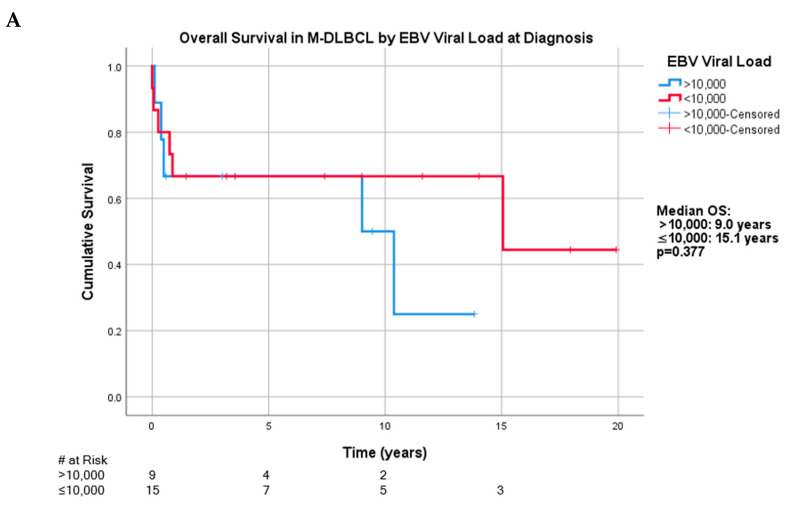
(**A**) Overall survival in the M-DLBCL-PTLD cohort with reported EBV quantification at the 10,000 copies/mL cutoff. (**B**) Overall survival in the M-DLBCL-PTLD cohort with reported EBV quantification at the 100,000 copies/mL cutoff.

**Table 1 cancers-13-00899-t001:** Clinical characteristics of post-transplant lymphoproliferative disorders (PTLD) population.

Variables	Sub-Types	Sample Size (*n*)	Percentage (%)
Total	-	66	100
Sex	Male	41	62.1
	Female	25	37.9
Race	White	34	51.5
	Black or African American	4	6.1
	Hispanic	23	34.8
	Asian	4	6.1
	Other	1	1.5
Transplant Age	Infant (<1 year old)	23	34.8
	Pediatric/Adolescent (1–18 years old)	22	33.3
	Adult (At least 18 years old)	21	31.8
Transplanted Organ	Heart	38	57.6
	Kidney	21	31.8
	Liver	4	6.1
	Multi-Organ	3	4.5
Time to PTLD Diagnosis	Early	6	9.1
	Late	38	57.6
	Very Late	22	33.3
ECOG Status	0–2	50	75.8
	3–4	16	24.2
CD20 Status	Positive	51	77.3
	Negative	10	15.1
	Unknown	5	7.6
Tumor EBER Status	Positive	49	74.2
	Negative	14	21.2
	Unknown	3	4.5
R-IPI Score	Low (0–2)	45	68.2
	High (3–4)	19	28.8
	Unknown	2	3.0

**Table 2 cancers-13-00899-t002:** Characteristics by histologic subtype of PTLD.

Category	All-Comers	ND-PTLD	P-PTLD	M-BL-PTLD	M-DLBCL-PTLD	M-PCN-PTLD	M-T/NK-PTLD	HL-PTLD
All	66	4	13	4	36	3	3	3
Transplant Age								
Infant	23	2	9	0	7	2	2	1
Pediatric/Adolescent	22	2	3	4	11	0	1	1
Adult	21	0	1	0	18	1	0	1
Transplanted Organ								
Heart	38	3	8	1	19	2	3	2
Kidney	21	1	3	3	13	0	0	1
Liver	4	0	1	0	3	0	0	0
Multi-Organ	3	0	1	0	1	1	0	0
Time to PTLD Diagnosis								
Early	6	0	1	0	4	1	0	0
Late	38	3	6	4	23	0	0	2
Very Late	22	1	6	0	9	2	3	1
First-Line Treatment								
Observation and RIS ± AV	18	2	5	1	5	1	2	2
Rituximab	10	2	2	0	5	1	0	0
R + CTX	30	0	4	2	24	0	0	0
CTX	8	0	2	1	2	1	1	1
ECOG Status								
0–2	50	4	9	4	27	3	1	2
3–4	16	0	4	0	9	0	2	1
Tumor EBER Status								
Positive	49	4	12	3	24	1	3	2
Negative	14	0	0	1	11	2	0	0
Unknown	3	0	1	0	1	0	0	1
EBV PCR Status								
Positive	50	3	11	3	27	1	3	2
Negative	6	0	1	0	4	1	0	0
Unknown	10	1	1	1	5	1	0	1
CD20 Status								
Positive	51	4	10	4	32	0	0	1
Negative	10	0	1	0	2	3	3	1
Unknown	5	0	2	0	2	0	0	1
R-IPI Score								
Low (0–2)	45	4	12	3	22	1	2	1
High (3–4)	19	0	1	1	13	1	1	2
Unknown	2	0	0	0	1	1	0	0

ND-PTLD: Non-destructive PTLD. P-PTLD: Polymorphic PTLD. M-DLBCL-PTLD: Monomorphic Diffuse Large B-Cell PTLD. M-PCN-PTLD: Monomorphic Plasma Cell Neoplasm-PTLD. M-T/NK-PTLD: Monomorphic T/Natural Killer-PTLD. HL-PTLD: Hodgkin’s Lymphoma PTLD.

**Table 3 cancers-13-00899-t003:** Treatment Strategy and Outcome for M-DLBCL-PTLD and P-PTLD Subtypes.

**First-Line Therapy for M-DLBCL-PTLD**	***N***	**Progressive Disease**	**Partial Response**	**Complete Response**
Observation or RIS ± AV	5	5	0	0
Rituximab	5	1	1	3
R + CTX	24	3	1	20
CTX	2	1	0	1
Total	36	10	2	24
**First-Line Therapy for P-PTLD**	***N***	**Progressive Disease**	**Partial Response**	**Complete Response**
RIS ± AV	5	0	0	5
Rituximab	2	0	0	2
R + CTX	4	1	0	3
CTX	2	0	0	2
Total	13	1	0	12

RIS ± AV: Reduction in Immunosuppressants with or without Antivirals; R + CTX: Rituximab and Chemotherapy; CTX: Chemotherapy Alone.

## Data Availability

The data presented in this study are available upon request from the authors.

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
