# Peer review of "Analysis of Post-Transplant Lymphoproliferative Disorder (PTLD) Outcomes with Epstein–Barr Virus (EBV) Assessments—A Single Tertiary Referral Center Experience and Review of Literature"

_cancers, 2021, doi:10.3390/cancers13040899_

Round 1

Reviewer 1 Report

Comments have been addressed properly.

The paper remains mainly confirmation and the heterogeneity bias cannot be resolved.

Author Response

Thank you for your time and reviewing the article.

Reviewer 2 Report

The manuscript has been improved by the authors. There is one sentnence that needs further clarification:

'Lastly, infants may have the longest time to primary EBV infection.'

Otherwise I'm happy for the article to be published.

Author Response

The manuscript has been improved by the authors. There is one sentence that needs further clarification:

'Lastly, infants may have the longest time to primary EBV infection.'

This has now been revised to:

"Lastly, infants may have a longer time to primary EBV infection than older patients, resulting in a delayed time to development of PTLD [1, 35]."

Reviewer 3 Report

Authors adequatley addressed the issues raised by the reviewer.

Author Response

Thank you for your time and reviewing the article.

This manuscript is a resubmission of an earlier submission. The following is a list of the peer review reports and author responses from that submission.

Round 1

Reviewer 1 Report

Lau et al report a single center retrospective study regarding the outcome of patients with PTLD. Findings are mainly confirmatory, and the heterogeneity of diagnoses and treatments represents a major limitation of this study.

Below are some comments:

  1. Given the very long follow-up, was histological diagnosis retrospectively confirmed by a pathologist?
  2. I suggest to include in the Table also the 3 patients with M-PTLD NK/T-cell type
  3. How long before diagnosis was EBV DNA measured? Please, provide median and range. Also, didn’t the method to measure EBV DNA change over time?

Reviewer 2 Report

Summary

This is a retrospective report of data regarding PTLD after solid organ transplant collected at one centre (Loma Linda University Medical Center (LLUMC)). They discuss 66 patients with a median follow-up time of 9.0 years (range between 0 and 24.7 years). Factors examined include age at transplant and diagnosis, type of organ transplant, subtype of PTLD, EBV load and the impact of these factors on response and survival. Overall, it’s a really interesting study done on data collated over many years. It is therefore important to publish this data. A limitation is that some comparisons do not have sufficient n numbers to be robust and some conclusions are overstated and should be toned down.  

Points to be addressed:

  1. I found this paragraph in the introduction difficult to follow. It would be useful to define pediatric versus children (or are they being used interchangeably) for the ease of the reader. Also, what does ‘unique interactions with immunosuppression and EBV infection’ mean.

‘Infants (under 1 year of age) may represent a unique subset of patients that develop PTLD, 95 potentially owing to unique interactions with immunosuppression and EBV infection. The incidence 96 of PTLD is higher in pediatrics than in adults due to primary EBV infection, and ranges from 1.2% to 97 30% depending on the transplanted organ [1,27–30]]. However, infants have a lower incidence of 98 PTLD than children (1 year to 10 years) [31]. Reasons for this finding are not fully elucidated. One 99 proposed mechanism is that maternal IgG antibodies are known to transfer before birth 100 transplacentally and are protective for bacterial and viral infections [32]. Furthermore, neonates are 101 relatively more tolerant of allografts and require less immunosuppression [33].

  1. Did the authors compare the time to PTLD diagnosis for the heart by age (infants versus pediatric or adult)? Figure 1B

  1. The numbers of patients is very low for liver (n= 4) and multi-organ (n=3) transplants to draw any firm conclusions about time to diagnosis and this should be stated. The data is much more robust for heart and kidney transplants.

  1. Again, I feel some of the conclusions/ analyses are on very low numbers of samples and the authors need to take care to point this out, or refer to literature if they want to stengthen their conclusion. For example below, when comparing EBER positivity %s in subsets that have 31 patients versus some subsets that only have 4 patients.

EBER-ISH was performed in 63 of 66 diagnoses. EBER-positive histology was observed in the 192 majority of patients (n=46/63, 73.0%). M-DLBCL-PTLD had the lowest level of EBER positivity 193 (n=20/31, 64.5%), compared to HL-PTLD (n= 6/6, 100%), P-PTLD (n=12/12, 100%), M-BL-PTLD (n=3/4, 194 75%) and ND-PTLD (n=5/7, 71.4%).

  1. The first statement in the conclusion shown below needs to be rephrased given there was no statistical significance to the finding.

‘In this single-center retrospective study, we found that infants had a numerically longer time to 343 PTLD diagnosis than older cohorts. Although the interaction p-value for age as a factor in time to 344 PTLD diagnosis did not reach statistical significance, there is a strong biologic rationale for this 345 finding which should be investigated in a larger cohort…’

Reviewer 3 Report

Lau et al. nicely presented characteristics and outcome of 66 patients after SOT developing a histology-proven PTLD. I only have a few comments:

1) Authors should include a flow chart for the pediatric and adult group of their patients including therapy and outcome. This will allow an understanding of the fate of the patients much better.

2) The methods section can be improved.

3) If possible, authors should also include patients of their center after HSCT developing PTLD as a separate group.